# The role of water molecules in the dissociation of an electron-molecule contact pair

Connor J. Clarke [1], E. Michi Burrow [1] & Jan R. R. Verlet [1,2] ✉

The hydrated electron, $e^-_{(aq)}$, is a potent reducing agent and a prototypical quantum solute. Reactions of $e^-_{(aq)}$ often involve a contact pair comprised of a molecule and electron that are hydrated within a single sphere. However, a molecular-level understanding of the solvent-driven coordinate that links the contact pair to the free dissociated $e^-_{(aq)}$ remains elusive. Here, we study this coordinate by kinetically trapping representative metastable intermediates as gas-phase clusters and probing them using photoelectron spectroscopy. We apply this methodology to uracil-water anion clusters, where key intermediates are identified with supporting quantum chemical calculations. Just a single water molecule drives the parent molecule and non-valence electron apart, thereby inhibiting geminate recombination to form the more stable valence-bound uracil anion. The electron-water binding is akin to bare water cluster anions, highlighting the link to larger clusters and $e^-_{(aq)}$. Our results provide a molecular-level view of quantum solute hydration and, more broadly, of how water-driven electron-transfer reactions proceed.

The hydrated electron, $e^-_{(aq)}$, plays a pivotal role in radiation chemistry and serves as the archetypical aqueous quantum solute, attracting much debate regarding its structure[1–4], non-adiabatic dynamics[5,6], and solvation at interfaces[4,7–11]. Photo-oxidation of an aqueous anion to form a molecule-electron contact pair[12], which is an intermediate towards $e^-_{(aq)}$, has been well-studied[13–17]. However, the critical subsequent step involving the dissociation of the contact pair to form $e^-_{(aq)}$ has been difficult to study directly, owing to the metastability of the contact pair. While computational studies have reproduced measured timescales of contact pair dissociation[13,18], experiments themselves offer limited mechanistic insight because of unavoidable competition from the thermodynamically favorable recombination channel. As the dissociation reaction is driven by the solvent, gaining a molecular-level view is at the heart of understanding the quantum hydration dynamics. Additionally, the reverse process underpins reactions of $e^-_{(aq)}$ with molecules, the rates of which have been studied for many decades[19].

We consider the uracil-electron reaction to experimentally explore the contact pair formation/dissociation coordinate. Uracil (U)

has received particular interest as an initial attachment site to induce genetic damage[20,21]. Reactions with $e^-_{(aq)}$ can generate nucleobase radical anions that then protonate to form damage-inducing dihydro-nucleobase products[22], or low-energy electrons can attach to nucleic acid bases to directly induce DNA/RNA strand breakages[23,24]. The initial reduction in these reactions is steered by the surrounding water molecules, making aqueous uracil, $U_{(aq)}$, an ideal system to study with biological relevance. However, as expected, $U^-_{(aq)}$ is far more stable than its contact pair, $[U:e^-]_{(aq)}$.

Here, we overcome the complications of the contact pair metastability and directly study structures that are far from equilibrium by exploiting non-valence anion-water clusters as a model for molecule-electron contact pairs. In this unique approach, geminate recombination can be inhibited through kinetic trapping, and the response of individual water molecules can be incremented to reveal key structures of the solvent-driven dissociation coordinate responsible for $e^-_{(aq)}$ formation.

In the gas phase, the anion $U^-$ forms as a dipole-bound state[25,26], where the excess electron is bound to the molecule in a non-valence

[1]Department of Chemistry, Durham University, Durham, United Kingdom. [2]J. Heyrovský Institute of Physical Chemistry, Czech Academy of Sciences, Prague 8, Czech Republic. ✉e-mail: j.r.r.verlet@durham.ac.uk

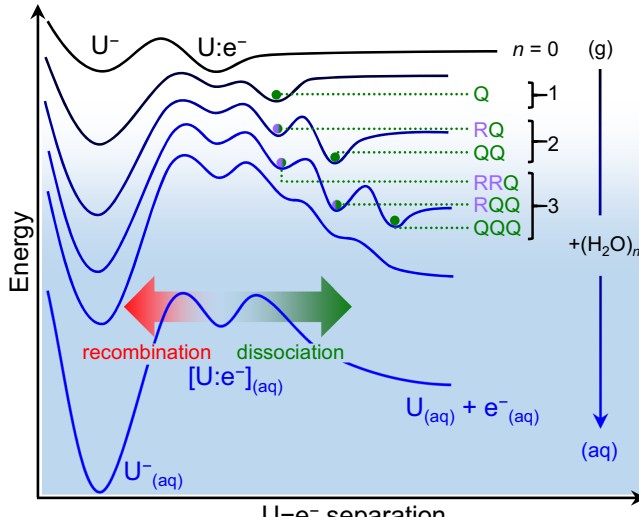

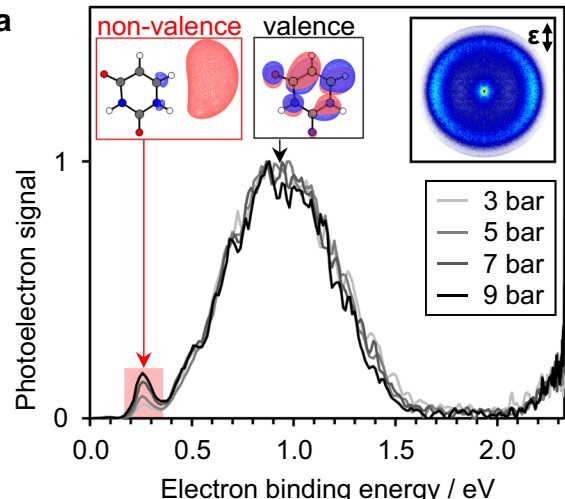

**Fig. 1 | Schematic of contact pair dissociation with increasing cluster size.** Schematic potential energy curve following the evolution of the non-valence dipole-bound state of uracil ($U{:}e^-$) into its aqueous analog, the contact pair $[U{:}e^-]_{(aq)}$, along a generalized $U{-}e^-$ separation coordinate. Rearrangement of the solvating water molecules can provoke either recombination into a valence anion $U^-$, or dissociation of the contact pair. Clusters of U with $n$ water molecules permit probing of the hydration-induced dissociation coordinate via kinetically trapped local minima, which are labeled according to their structural assignments in Fig. 3.

orbital[27,28]. This non-valence state is the gas-phase analog of $[U{:}e^-]_{(aq)}$ (see Fig. 1) and its hydration can be probed using anion-water clusters, $U^-(H_2O)_n$. However, past studies have demonstrated that the ground state of $U^-(H_2O)_n$ has valence character[21,25,29], preventing the formation of the microhydrated non-valence state. Here, we carefully control our source conditions such as to kinetically trap metastable non-valence states of $U^-(H_2O)_n$. By probing these delicate anion clusters with photoelectron spectroscopy, in conjunction with computational methods, the structures and stabilities of the non-valence states can be tracked with incremental hydration, $n$. Our results give insight into the role of individual water molecules in driving the dissociation of a contact pair into a precursor $e^-_{(aq)}$ state.

## Results

Figure 2a shows photoelectron spectra of $U^-(H_2O)_1$ acquired at $h\nu = 2.33$ eV for a range of backing gas pressures. The photoelectron signal is plotted against electron binding energy (equal to $h\nu$ minus the electron kinetic energy). There are two main features, which can be distinguished by their differing vertical detachment energy (VDE), defined as the electron binding energy at the peak maximum of the feature. The first feature is very broad, centered around VDE ≈ 1.0 eV: this spectrum has been observed before[21,25,29] and arises from photodetachment of valence-bound $U^-(H_2O)_1$ ($\pi_1^*$ state), where a single water molecule renders it as the lowest energy isomer. The spectral breadth arises from the disparate geometries between the initial (buckled) anion and the final (planar) neutral molecule. In addition, the photoelectron angular distribution (inset Fig. 2a) peaks perpendicular to the polarization vector of the light field (quantified by a negative anisotropy parameter, $\beta_2 < 0$), as expected for photodetachment from the $\pi_1^*$ molecular orbital[30,31]. The second feature in Fig. 2a is a sharper peak at VDE ≈ 0.25 eV. The narrow width suggests a small geometry change upon photodetachment and the photoelectron angular distribution is consistent with emission from a nodeless s-like orbital ($\beta_2 > 0$); this feature has the spectral signature for photodetachment from a non-valence state[32,33]. Detachment from a non-valence state of $U^-(H_2O)_1$ has not been previously observed. Figure 2a shows that the relative magnitude of the non-valence state feature increases with

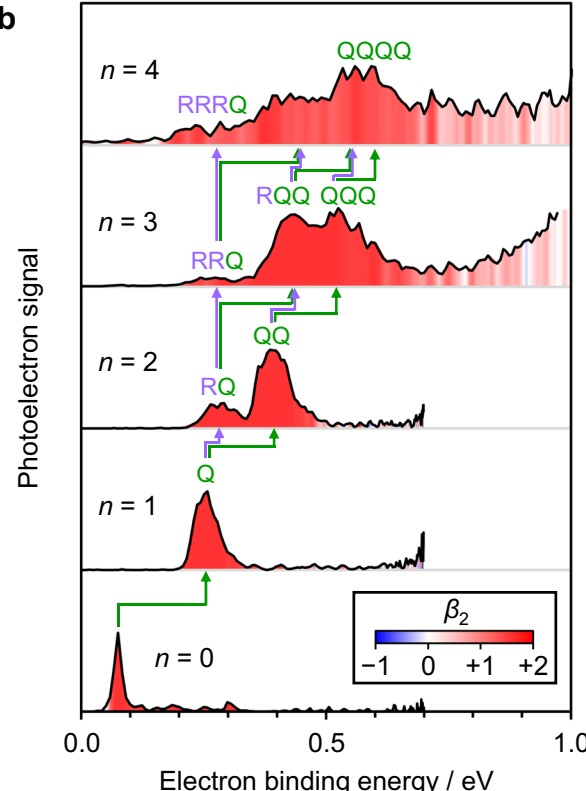

**Fig. 2 | Photoelectron spectra of $U^-(H_2O)_n$ isomers. a** Photoelectron spectrum of $U^-(H_2O)_1$ acquired at $h\nu = 2.33$ eV and at different backing gas pressures. Electronic structures of the anions responsible for each peak are shown inset, along with the corresponding photoelectron image at 9 bar Ar (with laser polarization vector indicated by $\varepsilon$). **b** Photoelectron spectra of $U^-(H_2O)_{0-4}$ acquired at lower photon energies, selectively detaching the non-valence state. Peaks are colored by the measured photoelectron anisotropy ($\beta_2$). Green and purple arrows highlight the increase in electron binding energy upon the addition of an electron-hydrating (Q) or ring-hydrating (R) water molecule, respectively. Source data are provided in this paper.

higher backing pressure, which in turn correlates with more efficient cooling in the molecular beam. Hence, we conclude that a larger fraction of non-valence anions are formed at lower cluster temperatures, indicating kinetic trapping of the non-valence state, $U^-(H_2O)_1$,

similar to observations of different isomers of $(H_2O)_n^-$ [34–39]. Non-valence states of $U^-(H_2O)_{n\leq4}$ could be kinetically trapped over the timescale required to perform the experiment (~200 μs).

Figure 2b displays photoelectron spectra of $U^-(H_2O)_{0-4}$ at photon energies: $hv = 0.70$ eV for $n = 0-2$, and $hv = 1.20$ eV for $n = 3, 4$. The use of lower photon energies enhances the photodetachment cross-section from the non-valence state [28,40,41], allowing us to discriminate against the valence state of $U^-(H_2O)_{1-4}$, which has a low photodetachment cross-section on account of the Wigner threshold law [42]. Peaks are labeled by their responsible isomeric structures, which are defined and assigned below.

The photoelectron spectrum of $U^-$ is narrow and anisotropic ($\beta_2 = +2.0$), corresponding to photodetachment from the dipole-bound state [25,43] with $VDE_0 = 75 \pm 6$ meV (where the subscript is used to indicate cluster size, $n$). For $U^-(H_2O)_1$, using $hv = 0.70$ eV, only a non-valence state is observed with $VDE_1 = 255 \pm 20$ meV. Between $VDE_0$ and $VDE_1$, the 180 meV increase in electron binding energy reflects the water-induced stabilization of the non-valence electron in $U^-(H_2O)_1$. While the differential stabilization is less than that for the valence-bound anion (~0.5 eV for the addition of one water molecule [44]), it is greater than the stabilization expected by combining the dipole moments of U and $H_2O$ [45]. Hence, the clustered water molecule is likely to be solvating the non-valence electron directly.

To explore the solvation by a single water molecule, we performed supporting quantum calculations. The binding sites for a water molecule in $U^-(H_2O)_1$, some of which have been identified in earlier

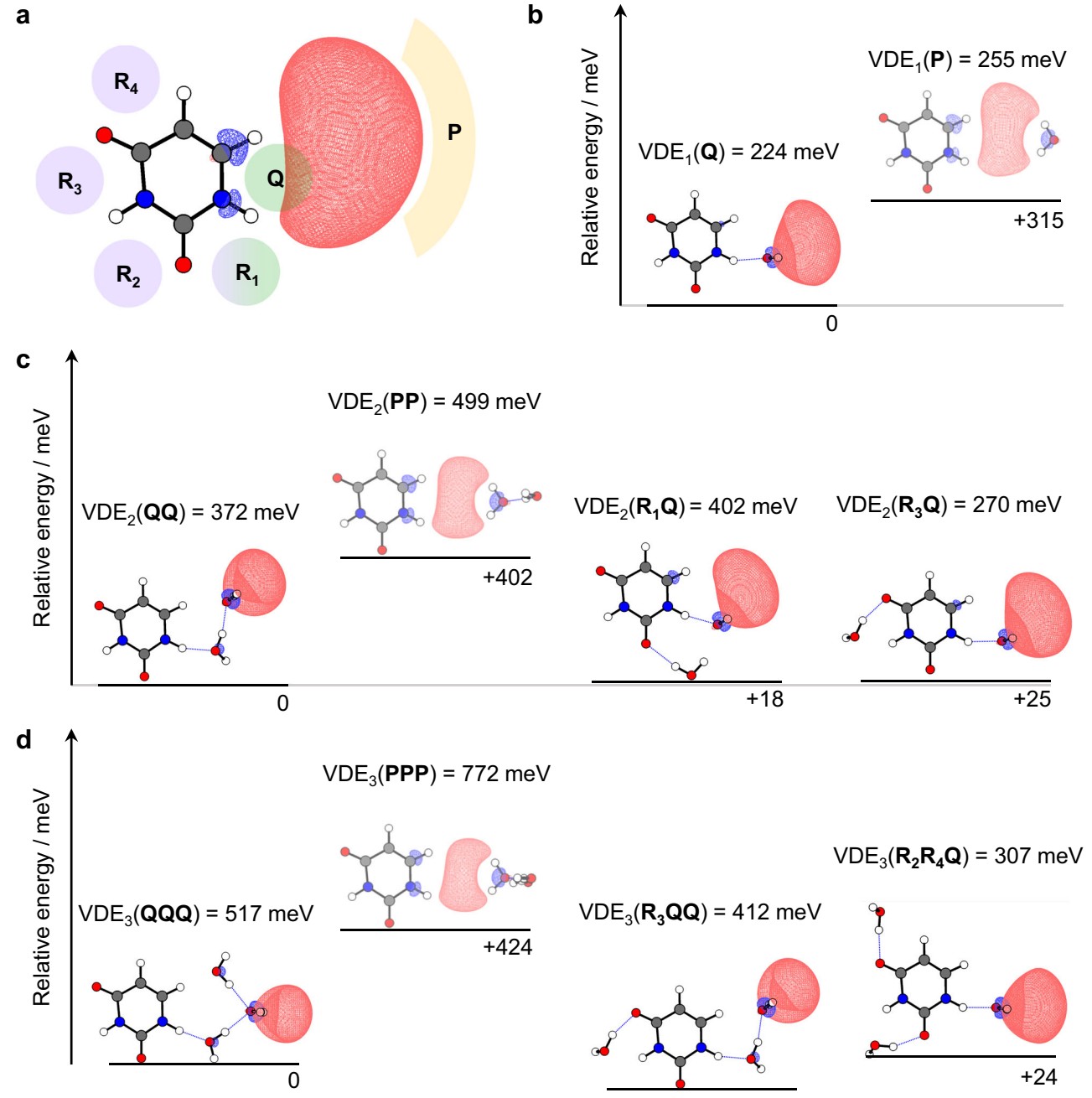

**Fig. 3 | Calculated structures of non-valence isomers of $U^-(H_2O)_{0-3}$. a** Electronic structure of the dipole-bound anion, $U^-$. Binding sites for clustered water molecules are labeled. Relative energies and VDEs, computed with CCSD(T)/aug-cc-pVDZ + 3s3p, of candidate optimized structures of $U^-(H_2O)_1$ (**b**), $U^-(H_2O)_2$ (**c**), and $U^-(H_2O)_3$ (**d**). Isovalues were chosen to be 75% of the maximum electron density of the non-valence state. Source data are provided in this paper.

computational studies[46], can be broadly grouped into three categories (as shown in Fig. 3a): sites that hydrate the nucleobase ring ($R_m$ where $m$ indicates the specific binding location); sites that bridge the void between U and the dipole-bound electron (Q); or sites at the outer periphery of the dipole-bound orbital (P). Relative energies and vertical detachment energies, $VDE_n(X)$ (where $X = R_m$, Q, or P), were computed as a direct link to experiments. In general, computed $VDE_n$ were found to be slightly lower than experimental values (Supplementary Table 3).

Unconstrained geometry optimizations for all $R_m$-isomers of $U^-(H_2O)_1$ converged to valence-bound anions. Some kinetically trapped non-valence isomers were found by constraining U to remain planar, as demonstrated in an earlier study[46], but this yielded $VDE_1(R_m) < 100$ meV, inconsistent with the observed $VDE_1 = 255 \pm 20$ meV. We thus conclude that R-isomers adiabatically convert to the valence-bound anion of $U^-(H_2O)_1$ within the timescale of the experiment. Exploration of Q- and P-isomers were then performed using fully relaxed geometry optimizations from an extensive set of starting geometries. Non-valence states were found in each case, with the two most stable structures for Q- and P-isomers shown in Fig. 3b. We found that $VDE_1(Q) = 224$ meV, suggesting that the Q-isomer is an appropriate candidate for the kinetically trapped non-valence anion observed for $U^-(H_2O)_1$. The P-isomer has $VDE_1(P) = 255$ meV, which is also consistent with the observed photoelectron signal. However, the Q-isomer is more stable than the P-isomer by 315 meV. Therefore, the Q-isomer, where the water molecule enhances the overall dipole moment but critically, also acts to hydrate the excess electron density, likely contributes most to the non-valence anion signal from $U^-(H_2O)_1$. In either case, we determine that the kinetically trapped non-valence anion of $U^-(H_2O)_1$ is stabilized by the water molecule hydrating the electron, rather than the uracil molecule. In Fig. 2b, we signify the stabilizing effect (180 meV) of the electron-hydrating water molecule by a green arrow.

The photoelectron spectrum of $U^-(H_2O)_2$ (Fig. 2b) shows two features, seemingly arising from different structural isomers. Both peaks are relatively narrow and the corresponding angular distributions are consistent with photodetachment from non-valence states[32,33]. The binding energies are $VDE_2 = 395 \pm 30$ meV and $278 \pm 30$ meV. A range of isomers are possible with permutations XY, where X, Y = $R_m$, Q, or P. All RR-isomers were found to converge to the valence-bound anion, consistent with the calculations on $U^-(H_2O)_1$. Hence, for each non-valence isomer, at least one of the water molecules must be at a site different to R. The difference between $VDE_1$ and $VDE_2$ is ~40 or ~20 meV for the two observed isomers of $U^-(H_2O)_2$ (green and purple arrows in Fig. 2b, respectively). Recalling that the electron-hydrating water increased the VDE by 180 meV in going from $U^-$ to $U^-(H_2O)_1$, we might anticipate that the isomer with the larger $VDE_2$ has two electron-hydrating water molecules (i.e., QQ, QP or PP), with the second water molecule offering similar, but slightly lower binding, as generally seen in clustering studies[47,48]. On the other hand, the isomer with the lower $VDE_2$ has a second water molecule that increases the VDE by a very small amount, which could be consistent with $U^-(H_2O)_1$ being hydrated at one of the R-sites (i.e., $R_m$Q- or $R_m$P-isomers).

We first consider the most likely structure of the isomer with $VDE_2 = 395 \pm 30$ meV. While fully relaxed geometry optimizations (from many different starting structures) led to a variety of structures, we show only the lowest energy structures of the candidate isomers. The lowest energy isomer is QQ, owing to the supporting hydrogen bonding network, as shown in Fig. 3c. The QP- and PP-isomers were 275 and 402 meV higher in energy, respectively, and the computed maxima in the photoelectron spectra for each isomer is predicted to be at $VDE_2(QQ) = 372$ meV, $VDE_2(QP) = 526$ meV, and $VDE_2(PP) = 499$ meV. Overall, the calculations suggest that the experimentally observed isomer with $VDE_2 = 395 \pm 30$ meV is the QQ-isomer, and thus, the non-

valence electron becomes more distant from the U molecule as it is preferentially hydrated from between the space dividing to the two. This conclusion also supports the previous supposition that the Q-isomer is the dominant non-valence isomer in $U^-(H_2O)_1$.

Explaining the origin of the photoelectron peak at $VDE_2 = 278$ meV for $U^-(H_2O)_2$ in Fig. 2b, we considered a range of $R_m$Q-isomers. The $R_2$Q-, $R_3$Q-, and $R_4$Q-isomers are 25, 25, and 103 meV higher in energy than the QQ-isomer, respectively. The corresponding VDEs range between 270–282 meV, such that all are close to the observed experimental $VDE_2 \approx 278$ meV. Therefore, isomers $R_2$Q, $R_3$Q, and $R_4$Q are candidate structures leading to this photoelectron signal. In contrast to $U^-(H_2O)_1$, the R-site water molecule does not induce the adiabatic formation of the valence anion of $U^-(H_2O)_2$ on the timescale of the experiment. Instead, it appears that the Q-site water molecule is 'locking' the excess electron into the non-valence state with the R-site water molecule simply hydrating neutral U, leading to the small additional increase in VDE (purple arrow in Fig. 2b). Since neutral U is planar and the excess electron is held at a distance, there is little driving force for the buckling motion that would otherwise encourage charge-transfer to U and the formation of the valence-bound anion. A schematic of the potential energy curve along the hydration-induced dissociation coordinate is shown in Fig. 1, where the RQ and QQ isomers are highlighted.

Finally, we note that the optimized structure describing $R_1$Q was calculated to have VDE = 402 meV, with a relative energy only 18 meV above the QQ structure. Therefore, the feature with $VDE_2 = 395 \pm 30$ meV could also contain contributions from the $R_1$Q-isomer, particularly on the edge of higher binding energy (where there is indeed a possible shoulder to the peak). However, as can be seen in Fig. 3c, the $R_1$-site water molecule also directly hydrates the excess electron and is therefore viewed more appropriately as intermediate to ring- and electron-hydrating, akin to a "solvent-shared" state that has been suggested to form during contact pair dissociation[17]. Hence, the potential presence of this isomer remains consistent with the idea that the excess electron is shifting further from the dipole-supporting U molecule with increasing hydration.

The photoelectron spectrum of $U^-(H_2O)_3$, shown in Fig. 2b, shows multiple distinct features that have become relatively broadened. Peaks are discernible at $VDE_3 = 530 \pm 50$ meV, $430 \pm 50$ meV, and $260 \pm 30$ meV. There is also some photoelectron signal at binding energies >0.7 eV, but this signal has starkly different angular distributions that can be correlated to electron emission from the valence-bound anion[30,31]. Applying a similar analysis for $U^-(H_2O)_3$ as we did for $U^-(H_2O)_2$, the feature at $VDE_3 = 530 \pm 50$ meV has increased by a further ~135 meV compared to the highest $VDE_2$ feature, suggesting it arises from a QQQ-like isomer (or $R_1$QQ). The feature at $VDE_3 = 430$ meV is likely an RQQ-isomer, and the feature at $VDE_3 = 260$ meV an RRQ-isomer.

The above prediction is consistent with the calculations. For the most stable computed QQQ structure, $VDE_3(QQQ) = 517$ meV, agreeing very well with the observed $VDE_3 = 530 \pm 50$ meV. Comparatively, the optimized PPP-isomer has a considerably higher $VDE_3(PPP) = 772$ meV, and is 424 meV higher in energy than the QQQ-isomer, providing further assurance that additional water molecules do not hydrate the excess electron from its outer periphery, but rather by nestling between the uracil molecule and the non-valence orbital (see Fig. 3d). The lowest energy RQQ structure was found to be $R_3$QQ, with $VDE_3(R_3QQ) = 412$ meV, in good agreement with the central feature at $VDE_3 = 430$ meV. But given the substantial width of the observed peak, other RQQ isomeric structures are likely to contribute too. Surprisingly, the $R_3$QQ-isomer was calculated to be lower in energy than the QQQ-isomer mentioned above, which may offer an explanation for the similar peak heights in the photoelectron spectrum of $U^-(H_2O)_3$ for both isomers. Finally, the feature at $VDE_3 = 260$ meV is expected to be an RRQ-isomer, where the single water molecule in the Q position

inhibits the formation of the valence-bound anion of U. For example, the non-valence state of the optimized $R_2R_4Q$ structure has $VDE_3(R_2R_4Q) = 307$ meV, in good agreement with the final observed peak. Once again, it appears that a single electron-hydrating water molecule is sufficient to prevent the excess electron from transferring onto the uracil ring (i.e., geminate recombination). Figure 1 again shows a schematic of how the observed structures drive the contact pair dissociation for $U^-(H_2O)_3$.

The photoelectron spectrum of $U^-(H_2O)_4$ is more complex, with several peaks partially resolved, and a much broader profile overall. The peaks at binding energies <0.7 eV can be correlated with non-valence states on account of their anisotropic angular distributions. Although there are too many isomers at this stage to confidently predict which peaks correlate with which isomers, the lowest binding energy edge of the photoelectron spectrum suggests the presence of RRRQ-isomers and the highest edge is likely attributable to QQQQ-isomers; the overall breadth of the signal suggests that several combinations of R and Q are intermediate. We were not able to generate any larger clusters with clear signals assignable to non-valence states, suggesting that the kinetically trapped species for $U^-(H_2O)_{n>4}$, if formed, had converted to valence anions on the timescale of our experiments.

## Discussion

Our results show that the addition of a single water molecule to the dipole-bound electron of $U^-$ drives the separation of the U molecule and the excess electron. The water molecule in the Q site binds by donating both its H-atoms to the excess electron distribution (see Fig. 3b). This binding motif is reminiscent of the binding of the excess electron in small water cluster anions, $(H_2O)_n^-$, the structures of which have been determined by IR action spectroscopy[49,50]. In these, a single water molecule in the cluster interacts closely with the non-valence electron through a similar double H-bond acceptor motif. For larger $(H_2O)_n^-$ clusters, the electron binding is enhanced, and the non-valence electron orbital becomes more confined[10,51,52], just as our calculations demonstrate for the all-Q isomers of $U^-(H_2O)_n$ (see Fig. 3). The connection between $U^-(H_2O)_n$ and $(H_2O)_n^-$ is further supported by considering the differential increase in binding with cluster size. In Fig. 4a, the VDE is plotted as a function of $n^{-1/3}$ (i.e., cluster size[47]) for the most stable non-valence isomers of $U^-(H_2O)_n$ and $(H_2O)_n^-$ [35,37], revealing similar gradients in the same range of VDE (~ 0.5 eV). The polar U offers additional stabilization of the non-valence electron (i.e., a vertical offset), but the similar gradients indicate that this effect is independent of $n$, suggesting that U has a minor influence on the electron binding as the degree of hydration is increased. This is consistent with the overall picture that additional water molecules hydrate from the Q site, moving the non-valence electron further away from U (see Supplementary Fig. 2 for calculated distances) and towards a structure akin to a cluster analog of $e^-_{(aq)}$, as schematically shown in Fig. 1. A recent computational study has predicted similar behavior, albeit for larger clusters: the dipole-bound anion of the model $H_3BNH_3$ molecule ($\mu = 5.356$ D) was subject to clustering of tens of water molecules and was found to localize the non-valence electron on the surface of the water cluster[53]. Our observation of all-Q isomers broadly supports this and suggests that a single water molecule is sufficient to instigate this transition.

From the perspective of the uracil-electron contact pair, ring-hydration (R-site) leads to geminate recombination of the non-valence electron with U, whereas hydration within the nucleobase-electron gap (Q-site) shifts the non-valence electron away from U and represents the hydration-driven dissociation coordinate, as sketched in Fig. 1. Recombination is energetically favored in the clusters formed here, as evidenced by the higher electron affinities of the valence state relative to the non-valence (all-Q) isomers, shown in Fig. 4b. However, the observed non-valence states were metastable on the timescale of the

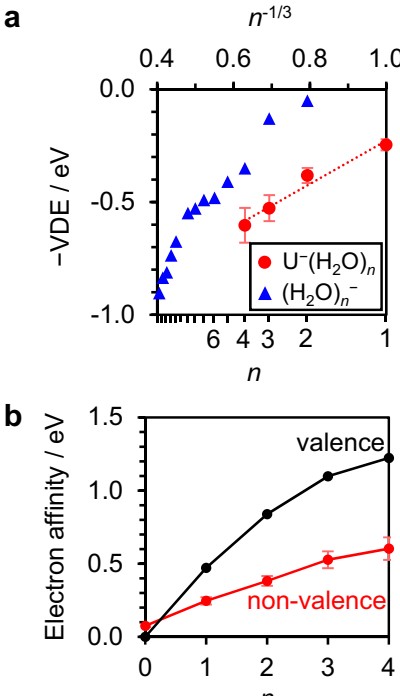

**Fig. 4 | Comparisons of electron binding energies of $U^-(H_2O)_n$. a** The vertical detachment energies (VDEs) of the most stable non-valence isomers of $U^-(H_2O)_n$ (red circles) compared against those of water cluster anions[35,50], $(H_2O)_n^-$ (blue triangles). A linear fit to the former is shown as a dotted line. **b** Electron affinities of $U(H_2O)_n$ associated with the valence (black) and non-valence anions (red). In both panels, error bars represent the standard deviations of Gaussian fits to the peaks. Source data are provided in this paper.

experiment (i.e., hundreds of microseconds). Since the valence state of $U^-(H_2O)_n$ is stabilized more with increasing hydration than the all-Q non-valence state, the energetic barrier separating the two states is generally expected to decrease (i.e., in a Marcus picture of charge-transfer[54]). This is consistent with our experimental observations, where we were unable to kinetically trap non-valence states of $U^-(H_2O)_{n>4}$. Overall, it appears that the energetic barriers between valence and non-valence states are largely dictated by hydration, which ultimately determines whether the nucleobase ring remains planar (as in non-valence states) or buckles into the minimum energy structure of the valence anion. Although our experiment identifies several likely intermediate structures that link the all-Q non-valence isomers to valence $U^-$, it does not probe the transition states between them.

Molecular clusters have long been utilized to study solvation effects with incremental detail in a bottom-up approach, taking advantage of the most interrogative spectroscopic techniques available in the gas phase[47,55,56]. The cluster approach has been particularly successful for hydrated electrons, accurately extrapolating excited state lifetimes[5,6] and binding energies[35,37,57,58]. In addition to this, we exploited the propensity for cold anion-solvent clusters to form kinetically trapped isomers, thus isolating metastable states that are not distinctly observable in a bulk environment. By directly probing the electronic structure of non-valence states of $U^-(H_2O)_n$, we identify key water-binding motifs that govern the distance between the nucleobase and the excess electron.

Based on the cluster structures determined here, the initial step in the dissociation of a contact pair appears to be the incursion of a water molecule between the molecule and the excess electron, to which further water molecules push in and drive the electron further from

the parent into $e^-_{(aq)}$. Although comparisons between cluster and bulk phenomena should be cautioned, this hydration-induced mechanism is in accord with earlier suggestions of a bulk diffusion-controlled dissociation, where the initial motion of a single water molecule is key[13,59]. On the other hand, geminate recombination into a valence anion can be promoted by the response of several water molecules surrounding the parent molecule. Our use of molecule-water cluster anions offers a new route to probing the molecular structures along the hydration coordinate that determines the fate of the non-valence electron in a contact pair and the creation of $e^-_{(aq)}$, and highlights the instrumental role played by individual water molecules along the reactive uracil-electron coordinate.

## Methods

### Experimental

A supersonic expansion of $U^-(H_2O)_n$ cluster anions was generated by heating U to 220 °C in a pulsed Even-Lavie valve[60] that was backed with high-pressure inert gas (Ar flowed over $H_2O$), and then attaching electrons produced from a hot filament[61]. Ions were mass-separated using time-of-flight mass spectrometry and were exposed to light at the center of a velocity map imaging photoelectron spectrometer to produce photoelectron images from which photoelectron spectra (and angular distributions) were obtained using the MELEXIR algorithm[62]. Excitation light was produced using an Nd:YAG-pumped optical parametric oscillator (or the second harmonic of the fundamental output from the Nd:YAG laser). Photoelectron images were calibrated using the iodide anion and have a resolution of ~3% of the electron's kinetic energy. A consideration of the formation processes of non-valence $U^-(H_2O)_n$ cluster anions is given in the Supplementary Information.

### Computational

The optimized structure of each isomer was found using density functional theory (CAM-B3LYP/aug-cc-pVDZ + 3s3p)[63,64], initiated from a wide range of configurations. The basis set contains extra diffuse functions on the C and N atoms closest to the positive pole of U[46]. Relative energies and vertical detachment energies $VDE_n(X)$ (where $X = R_m$, Q, or P) were computed with greater accuracy using ab initio CCSD(T)/aug-cc-pVDZ + 3s3p[65]. Further benchmarking details, as well as structures and electron-molecule distances, are available in the Supplementary Information. All calculations were conducted with Gaussian 16[66].

### Reporting summary

Further information on research design is available in the Nature Portfolio Reporting Summary linked to this article.

## Data availability

The raw and displayed data generated in this study have been deposited in the Zenodo database under the accession code https://doi.org/10.5281/zenodo.14710932.

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

## Acknowledgements

This work was funded by the EPSRC under grant EP/V007971/1 (E.M.B., J.R.R.V.), the OP JAK project No. CZ.02.01.01/00/22_008/0004649 (QUEENTEC) (J.R.R.V.), and Durham University for a Scholarship (C.J.C.). For the purpose of open access, the authors have applied a Creative Commons Attribution (CC BY) license to any Author Accepted Manuscript version arising.

## Author contributions

J.R.R.V. conceived the overall project. J.R.R.V., C.J.C., and E.M.B. developed the experimental methodology. C.J.C. and E.M.B. performed the experiments and interpreted the data. C.J.C. provided theoretical calculations and performed further data analysis. C.J.C. wrote the manuscript with input from J.R.R.V. and E.M.B.

## Competing interests

The authors declare no competing interests.
