## [Transparent Peer Review file · Nature Communications]

The role of water molecules in the dissociation of an electron-molecule contact pair

Corresponding Author: Professor Jan Verlet

Version 0:

Reviewer comments:

Reviewer #1

(Remarks to the Author)

The authors report a groundbreaking study of metastable dipole bound uracil-water anion clusters using photoelectron spectroscopy and DFT quantum calculations.

While previous work showed that dipole bound anion uracil water structures existed. This work breaks much new ground and substantially adds to our understanding of these systems. There are a number of notable findings: 1. that a single water can insert between the dipole bound electron and uracil, 2. That subsequent waters can solvate the electron or the uracil. 3. That dipole bound states can be distinguished from valence states by use of photoelectron anisotropy.

The experimental methods are at the forefront of this area. Especially significant is the use of photoelectron anisotropy as mentioned. The theoretical treatments for dipole bound states requires the use a diffuse DFT basis sets because weak bonding (0 to 0.5 eV) results in a loosely bound excess electron with substantial diffuse character. The authors employ CCSD(T) with a constructed aug-cc-pVDZ+3s3p basis set which they show in their supplemental information provides excellent agreement with experimental VDEs. This is a very positive feature of this work as it gives their calculations and assignments credibility.

Points for authors to consider:

1. The authors should make clear in the supplemental the coefficients used in aug-cc-pVDZ+3s3p so it could be employed by other researchers.
2. The authors use the term solvation often when additional waters are added to the cluster. Hydration is a more appropriate term. Along the same lines on page 10 the comparison of the QQ and QR isomers to a CTTS state in aqueous solution is problematic since dipole bound states are energetically suppressed in solution.

Reviewer #2

(Remarks to the Author)

This is a very well written paper dealing with a topic of broad interest and coming from a team that knows the field well. I read the paper thoroughly and was quite interested in the "story" it tells. I am supportive of publishing it, but I would like the authors to address a few minor points first.

1. I could not find the early very nice overview about the solvated electron by Jim Coe cited anywhere. I think it would be appropriate and fair to add some discussion of that worker's pioneering work on this topic.
 2. I will refer to Fig. 1 to explain this question. The paper clearly shows how the authors' experiments can probe the relative stabilities and dynamic interconversions that connect the state in which the electron is bound to a Q-site toward one in which water molecules move away from the electron toward the Uracil's polar sites (forming R sites). However, what I remain a bit confused about is (a) whether the authors can say anything about the rates of these Q-to-R changes, and (b) whether they can say anything about the rates at which the electron ends up in a valence orbital of the U- anion.
 3. Finally, I don't see anything in this report that explains to us the pathway by which the electron moves from a non-valence (i.e., dipole-bound) orbital sitting largely on an OHH water molecule, inward into a non-valence orbital sitting between the U molecule's dipole and an inward-pointing HHO water, and finally (c) "upward" out of the non-valence region into the π^* orbital of the U. I assume the experimental data just don't provide information about this, correct?
- Once these issues are resolved, I fully support publishing this very nice report.

Reviewer #3

(Remarks to the Author)

This is a remarkable paper in which the authors have discovered a new series of $U(H_2O)_n$ - cluster anions in which the excess electron is non-covalently (and presumably dipole-)bound. It has been accepted for close to 30 years that the addition of one water molecule to a uracil anion transforms the electron binding motif from dipole- to valence-bound. Here, the authors have found experimental conditions in which (a) non-covalent states for hydrated uracil anion are seen for the first time and (b) dominate the photoelectron spectrum. These observations are sufficiently novel to warrant publication in Nature Comm.

My main issue with the manuscript is that it overemphasizes the connection to electron solvation dynamics and underplays the cluster aspect of the work. Everyone involved with water cluster anions and related species has learned that it is a bit risky to extrapolate to bulk phenomena from the properties of very small clusters, and it's unclear that this work, as nice as it is, is relevant to the dynamics of contact-pair formation in solution. I think it's fair to use this as motivation, but I recommend that the authors downplay this aspect of the work and say a bit more about its importance in the field of cluster science.

Version 1:

Reviewer comments:

Reviewer #2

(Remarks to the Author)

The authors did address in a satisfactory manner all of the issues raised by the three reviewers.

REVIEWER COMMENTS

Reviewer #1 (Remarks to the Author):

The authors report a groundbreaking study of metastable dipole bound uracil-water anion clusters using photoelectron spectroscopy and DFT quantum calculations. While previous work showed that dipole bound anion uracil water structures existed. This work breaks much new ground and substantially adds to our understanding of these systems. There are a number of notable findings: 1. that a single water can insert between the dipole bound electron and uracil, 2. That subsequent waters can solvate the electron or the uracil. 3. That dipole bound states can be distinguished from valence states by use of photoelectron anisotropy.

The experimental methods are at the forefront of this area. Especially significant is the use of photoelectron anisotropy as mentioned. The theoretical treatments for dipole bound states requires the use a diffuse DFT basis sets because weak bonding (0 to 0.5 eV) results in a loosely bound excess electron with substantial diffuse character. The authors employ CCSD(T)) with a constructed aug-cc-pVDZ+3s3p basis set which they show in their supplemental information provides excellent agreement with experimental VDEs. This is a very positive feature of this work as it gives their calculations and assignments credibility.

Points for authors to consider:

1. The authors should make clear in the supplemental the coefficients used in aug-cc-pVDZ+3s3p so it could be employed by other researchers.

This has now been added to the Supporting Information.

2. The authors use the term solvation often when additional waters are added to the cluster. Hydration is a more appropriate term.

Agreed – we have applied the change.

Along the same lines on page 10 the comparison of the QQ and QR isomers to a CTTS state in aqueous solution is problematic since dipole bound states are energetically suppressed in solution.

We have adjusted our original comparison to a 'solvent-shared' state that is expected to take place during CTTS (and contact pair) dissociation. The main purpose of the comparison was to highlight that the R₁-hydration is more akin to Q-hydration than R₂₋₄-hydration. Instead, we place more emphasis on this point, but also guide to

reader to the relevant citation showing the 'solvent-shared' state (to illustrate that this class of structure has precedent).

Reviewer #2 (Remarks to the Author):

This is a very well written paper dealing with a topic of broad interest and coming from a team that knows the field well. I read the paper thoroughly and was quite interested in the "story" it tells. I am supportive of publishing it, but I would like the authors to address a few minor points first.

1. I could not find the early very nice overview about the solvated electron by Jim Coe cited anywhere. I think it would be appropriate and fair to add some discussion of that worker's pioneering work on this topic.

We thank the reviewer for the suggestion and have now included key citations to Coe (and others), and have made some broader comments about cluster work on the hydrated electron and its extrapolation to the bulk.

2. I will refer to Fig. 1 to explain this question. The paper clearly shows how the authors' experiments can probe the relative stabilities and dynamic interconversions that connect the state in which the electron is bound to a Q-site toward one in which water molecules move away from the electron toward the Uracil's polar sites (forming R sites). However, what I remain a bit confused about is (a) whether the authors can say anything about the rates of these Q-to-R changes, and (b) whether they can say anything about the rates at which the electron ends up in a valence orbital of the U⁻ anion.

We are not able to obtain rates from our experiment. We can only say that the metastable states (that are separated by barriers) are metastable on the timescale of the experiment (i.e. they can be observed), which is on the order of hundreds of microseconds (as noted in the manuscript). We cannot change the "temperature" of the clusters and indeed, we do not know the internal temperature – only that we are "cold" (on the order of 10s K). Unfortunately, therefore, we are unable to provide further insight. We know that the valence states of $U^-(H_2O)_{n \geq 1}$ are always the lowest energy, but the interconversion rate is not probed by our experiment. We have added a brief comment in the discussion / SI concerning interconversion rates.

3. Finally, I don't see anything in this report that explains to us the pathway by which the electron moves from a non-valence (i.e., dipole-bound) orbital sitting largely on an OHH water molecule, inward into a non-valence orbital sitting between the U

molecule's dipole and an inward-pointing HHO water, and finally (c) "upward" out of the non-valence region into the π^* orbital of the U. I assume the experimental data just don't provide information about this, correct?

We cannot obtain information about the transition states between the various structures and so cannot obtain the complete pathway. Nonetheless, our work offers information about likely intermediates along the reaction coordinate as described by the reviewer. In particular, it is clear that the final step from non-valence to valence involved an out-of-plane buckling of the uracil ring – a transition we have also previously shed some light on using $U^-(Ar)_n$ clusters (Phys. Chem. Chem. Phys., 2024, **26**, 20037–20045).

We have added comments to the discussion outlining this limitation.

Once these issues are resolved, I fully support publishing this very nice report.

Reviewer #3 (Remarks to the Author):

This is a remarkable paper in which the authors have discovered a new series of $U(H_2O)_n$ - cluster anions in which the excess electron is non-covalently (and presumably dipole-)bound. It has been accepted for close to 30 years that the addition of one water molecule to a uracil anion transforms the electron binding motif from dipole- to valence-bound. Here, the authors have found experimental conditions in which (a) non-covalent states for hydrated uracil anion are seen for the first time and (b) dominate the photoelectron spectrum. These observations are sufficiently novel to warrant publication in Nature Comm.

My main issue with the manuscript is that it overemphasizes the connection to electron solvation dynamics and underplays the cluster aspect of the work. Everyone involved with water cluster anions and related species has learned that it is a bit risky to extrapolate to bulk phenomena from the properties of very small clusters, and it's unclear that this work, as nice as it is, is relevant to the dynamics of contact-pair formation in solution. I think it's fair to use this as motivation, but I recommend that the authors downplay this aspect of the work and say a bit more about its importance in the field of cluster science.

While we wholeheartedly agree with the "risk" in extrapolating to bulk phenomena, we also do note how well hydrated electron work has fared in this context. Nevertheless, we agree that there is a long way between these relatively small clusters and the bulk solvent. We have added further discussion to highlight the importance of observing metastable solvent-solute species from the cluster perspective and outline how these clusters enable the electronic character to be

probed in detail, which would not be possible in a bulk environment. We also emphasize that caution should be applied when comparing cluster phenomena to bulk phase behaviour, and have reworded parts of the manuscript to place more focus on the cluster aspect.